# Single-nucleus transcriptomics reveal the cytological mechanism of conjugated linoleic acids in regulating intramuscular fat deposition

**Liyi Wang**[1,2,3], **Shiqi Liu**[1,2,3], **Shu Zhang**[1,2,3], **Yizhen Wang**[1,2,3], **Yanbing Zhou**[1,2,3]*, **Tizhong Shan**[1,2,3]*

[1]College of Animal Sciences, Zhejiang University, Hangzhou, China; [2]Key Laboratory of Molecular Animal Nutrition, Zhejiang University, Hangzhou, China; [3]Key Laboratory of Animal Feed and Nutrition of Zhejiang Province, Hangzhou, China

*For correspondence: zhouyanbing@zju.edu.cn (YZ); tzshan@zju.edu.cn (TS)

**Competing interest:** The authors declare that no competing interests exist.

## eLife Assessment

This study provides **valuable** information on the single nucleus RNA sequencing transcriptome, pathways, and cell types in pig skeletal muscle in response to conjugated linoleic acid (CLA) supplementation. Based on the comprehensive data analyses, the data are considered **compelling** and provide new insight into the mechanisms underlying intramuscular fat deposition and muscle fiber remodeling. The study contributes significantly to the understanding of nutritional strategies for fat infiltration in pig muscle.

**Abstract** Conjugated linoleic acids (CLAs) can serve as a nutritional intervention to regulate quality, function, and fat infiltration in skeletal muscles, but the specific cytological mechanisms remain unknown. Here, we applied single-nucleus RNA-sequencing (snRNA-seq) to characterize the cytological mechanism of CLAs regulates fat infiltration in skeletal muscles based on pig models. We investigated the regulatory effects of CLAs on cell populations and molecular characteristics in pig muscles and found CLAs could promote the transformation of fast glycolytic myofibers into slow oxidative myofibers. We also observed three subpopulations including SCD$^+$/DGAT2$^+$, FABP5$^+$/SIAH1$^+$, and PDE4D$^+$/PDE7B$^+$ subclusters in adipocytes and CLAs could increase the percentage of SCD$^+$/DGAT2$^+$ adipocytes. RNA velocity analysis showed FABP5$^+$/SIAH1$^+$ and PDE4D$^+$/PDE7B$^+$ adipocytes could differentiate into SCD$^+$/DGAT2$^+$ adipocytes. We further verified the differentiated trajectory of mature adipocytes and identified PDE4D$^+$/PDE7B$^+$ adipocytes could differentiate into SCD$^+$/DGAT2$^+$ and FABP5$^+$/SIAH1$^+$ adipocytes by using high intramuscular fat (IMF) content Laiwu pig models. The cell-cell communication analysis identified the interaction network between adipocytes and other subclusters such as fibro/adipogenic progenitors (FAPs). Pseudotemporal trajectory analysis and RNA velocity analysis also showed FAPs could differentiate into PDE4D$^+$/PDE7B$^+$ preadipocytes and we discovered the differentiated trajectory of preadipocytes into mature adipocytes. Besides, we found CLAs could promote FAPs differentiate into SCD$^+$/DGAT2$^+$ adipocytes via inhibiting c-Jun N-terminal kinase (JNK) signaling pathway in vitro. This study provides a foundation for regulating fat infiltration in skeletal muscles by using nutritional strategies and provides potential opportunities to serve pig as an animal model to study human fat infiltrated diseases.

## Introduction

Meat is one of the most important sources of animal protein for humans, and its quality is associated with human health. Recently, due to the development of economic levels and the improvement of living standards, people are seeking for 'less but better' meat (*Sahlin and Trewern, 2022*). Intramuscular fat (IMF) deposition is a key factor positively related to meat quality traits and lipo-nutritional values of meat, such as flavor, tenderness, and juiciness (*Hausman et al., 2014*; *Yi et al., 2023*). Besides, fat infiltration in skeletal muscle (also known as myosteatosis) is the pathologic fat accumulation in skeletal muscle with poor metabolic and musculoskeletal health; it always accompanied by the decline of muscle quality and function (*Biltz et al., 2020*; *Jiang et al., 2019*). Myosteatosis is now considered as a common feature of aging and is also related to some diseases (*Wang et al., 2024*). However, the occurrence mechanism and cell sources of fat accumulation in skeletal muscle is very complicated. Recently, with the rapid development of multi-omics including single-cell RNA sequencing (*Tabula Muris Consortium et al., 2018*), single-nucleus RNA-seq (snRNA-seq) (*Petrany et al., 2020*), and spatial transcriptomics (ST) (*Jin et al., 2021*), more and more cell types have been found to contribute to lipid deposition in skeletal muscle including myogenic cells, e.g., satellite cells (SCs) (*Asakura et al., 2001*) and myogenic factor 5 (Myf5)[+] mesenchymal stem cells (MSCs) (*Yin et al., 2013*) and non-myogenic cells, e.g., fibro/adipogenic progenitors (FAPs) (*Uezumi et al., 2010*), fibroblasts, myeloid-derived cells (*Xu et al., 2021*), pericytes (*Farrington-Rock et al., 2004*), endothelial cells (ECs) (*Lang et al., 2008*), PW1[+]/Pax7[-] interstitial cells (PICs) (*Mitchell et al., 2010*), and side population cells (SPs) (*Tamaki et al., 2002*) based on animal models. Hence, more and more researches have been focusing on exploring the regulatory mechanism of myosteatosis at the cytological levels. Besides, fat infiltration in skeletal muscle is regulated by many influential triggers, including aging, metabolic and nonmetabolic diseases, disuse and inactivity, and muscle injury (*Wang et al., 2024*). Many genes and signaling pathways participate in the formation and regulation of fat infiltration in skeletal muscle (*Biferali et al., 2021*; *Woscyzna et al., 2021*). However, the specific mechanism of nutrients regulates fat infiltration in muscle remains unknown.

Nutritional regulation strategy is one of the most vital strategies to regulate lipid accumulation in skeletal muscle based on some animal models and clinic trials, including vitamins (*Gilsanz et al., 2010*; *Zhao et al., 2020*), conjugated linoleic acid (CLAs) (*van Vliet et al., 2020*), linseed (*Wei et al., 2016*), plant extract (*You et al., 2023*), and so on. Hence, exploring the potential nutritional strategies to regulate fat accumulation in skeletal muscle is valuable for animal production and human health. Pigs are not only an important source of animal protein in the human diet but also serve as a valuable model for human medical biology due to their similar physiological structure in terms of size, metabolic characteristics, and cardiovascular system (*Groenen et al., 2012*; *Lunney et al., 2021*). Chinese local pig species could be used as excellent animal models to study the mechanism of lipid deposition due to the fact that they have better meat quality and high IMF content. A high IMF content always results in better juiciness, tenderness, and flavor of pork. There are different myogenesis potential in neonatal skeletal muscle between Laiwu pigs and Duroc pigs (*Xu et al., 2023*). Our previous study has revealed the cell heterogeneity and transcriptional dynamics of lipid deposition in skeletal muscle of Laiwu pigs and we found high IMF content Laiwu pigs had lower muscle fiber diameter (*Wang et al., 2023b*). Heigai pig is a model of Chinese indigenous pig breeds, which has advantages including high farrowing rate, good pork quality, and strong disease resistance (*Wang et al., 2022a*). Specially, we have found CLAs can improve meat quality, especially increase IMF content in both lean type pig breeds and fat type pig breeds (*Wang et al., 2021*; *Wang et al., 2022b*). However, although many studies have discussed the effects of CLA on IMF deposition, there are still gaps in the cytological mechanism of CLAs in regulating lipid deposition in skeletal muscle.

Here, we present an snRNA-seq dataset collected from *longissimus dorsi muscle* (LDM) of Heigai pigs after feeding CLAs supplement to allow for analyzing heterogeneity of transcriptional states in muscles. We investigated the regulatory effects of CLAs on the muscle fiber-type transformation and IMF deposition. We also identified the differentiation trajectories of three subclusters in adipocytes based on Heigai pig models and high IMF content Laiwu pig models. Based on the pseudotemporal trajectories analysis, we found CLAs could promote FAPs differentiate into SCD[+]/DGAT2[+] adipocytes via regulating mitogen-activated protein kinase (MAPK) signaling pathway. This study paves a way to regulate lipid accumulation in muscles by using nutritional strategies and provides theoretical basis on using pig as an animal model to study human muscle-related diseases.

## Results

### CLAs changed cell populations and transcriptional dynamics in LDM

Our previous study discovered CLAs improved IMF content in LDM of Heigai pigs (*Wang et al., 2022b*), here, we also found CLAs significantly increased TG content but significantly decreased TC content in LDM of pigs (*Figure 1A*). Meanwhile, immunofluorescence staining results showed more lipid droplets in the LDM of the CLAs group (*Figure 1B*). To investigate the changes in cell heterogeneity in pig muscles after CLAs treatment at the cellular level, we performed snRNA-seq from LDM tissue of Heigai pigs using the 10x Genomics Chromium platform (*Figure 1C*). First, after Cell Ranger analyses, the estimated number of cells, fraction of reads in cells, mean reads per cell, median genes per cell, and median UMI counts per cell in LDM were shown in *Figure 1—figure supplement 1A*. We obtained 25,507 cells from two individual libraries, comprising 10,835 cells from CON group and 11,705 cells from CLA group for the downstream analysis after the quality control of snRNA-seq data (*Figure 1—figure supplement 1B*). Based on the Seurat package, we used Uniform Manifold Approximation and Projection (UMAP) plots to show the different subclusters (*Figure 1D*). We identified eight different clusters in two groups of pig muscles, including myofibers (*CAPN3*), FAPs/fibroblasts (*PDGFRA*), ECs (*CD34*), adipocytes (*PPARG*), immune cells (*PTPRC*), muscle SCs (MuSCs) (*PAX7*), myeloid-derived cells (*MRC1*), and pericytes (*RGS5*) using the expression of marker genes (*Figure 1E*). Next, the percentage of these cell types showed differences in different groups (*Figure 1F*). Compared with the CON group, the FAPs/fibroblasts (3.39% vs 8.31%), ECs (1.26% *vs* 2.94%), adipocytes (1.74% vs 2.37%), myeloid-derived cells (0.93% vs 2.17%), and pericytes (0.45% vs 0.7%) had a higher proportion in the CLA group. However, CLA decreased the proportion of myofibers (90.46% vs 81.1%). The top 10 most differentially expressed genes (DEGs) were shown in the heatmap and the bar plot showed the top 3 KEGG enrichment of DEGs between the eight different cell types in LDM (*Figure 1G*). Besides, violin plots displayed CLAs upregulated the expression of mature adipocyte master genes including *ADIPOQ*, *FABP4*, *PLIN1*, and *LIPE*, adipogenic marker genes including *PPARG*, *PPARA*, *CEBPA*, and *CEBPB*, and lipid metabolism-related genes including *LPL*, *ELOVL4*, *ACAA2*, and *HACD2* in pig muscles (*Figure 1—figure supplement 1C–E*). These results indicated the cell types in LDM of Heigai pigs had significant differences after feeding CLAs which might induce the alterations in lipid deposition.

### Characterization of myofibers after CLAs treatment through clustering analysis

To explore the changes in myofibers after CLA treatment, we next carried out a subcluster analysis and investigated the cell heterogeneity of myofibers. UMAP plots displayed the distribution in different subsets of myofibers (*Figure 2A*). Based on our previous study (*Wang et al., 2023b*) and different gene expression in myofibers, we also characterized 6 different cell types in myofibers, including I myofibers (*MYH7*), IIA myofibers (*MYH2*), IIX myofibers (*MYH1*), IIB myofibers (*MYH4*), myotendinous junctions (MTJ, *ANKRD1*), and neuromuscular junction (NMJ, *ABLIM2*) (*Figure 2B*). The bar plot showed the proportion of type I myofibers (17.54% vs 22.01%) and IIA myofibers (3.64% vs 7.52%) had an increased tendency while the proportion of IIX myofibers (10.09% vs 3.87%), IIB myofibers (63.88% *vs* 59.35%), MTJ (6.41% vs 0.28%), and NMJ (5.32% *vs* 0.09%) had a reduced tendency in CLA group (*Figure 2C*). Besides, violin plot displayed CLAs increased the expression of myofiber-type marker genes (*MYH7*, *MYH2*, *MYH1*, and *MYH4*), myofiber-type transformation-related genes (*PPARGC1A*, *STK11*, and *HDAC1*), and oxidation-related genes (*COX5A*, *COX5B*, and *COX8A*) but decreased glycolysis-related genes (*PFKM*, *HK2*, and *LDHC*) (*Figure 2D*). Additionally, the expression of *MYHCI* and *COX5B* in LDM of Heigai pigs was also significantly increased after CLA treatment (*Figure 2E*). The top 10 DEGs between the six cell types in myofibers were shown in heatmap (*Figure 2F*). Functional enrichment analyses revealed the enrichment of metabolic pathways, oxidative phosphorylation, and thermogenesis in I myofibers and calcium, cGMP-PKG, and MAPK signaling pathway in IIB myofibers by using KEGG pathways (*Figure 2G*). These data discovered the significantly heterogeneity in myofibers between two groups and CLAs could promote slow oxidative myofibers switch into fast glycolic myofibers in pig muscles.

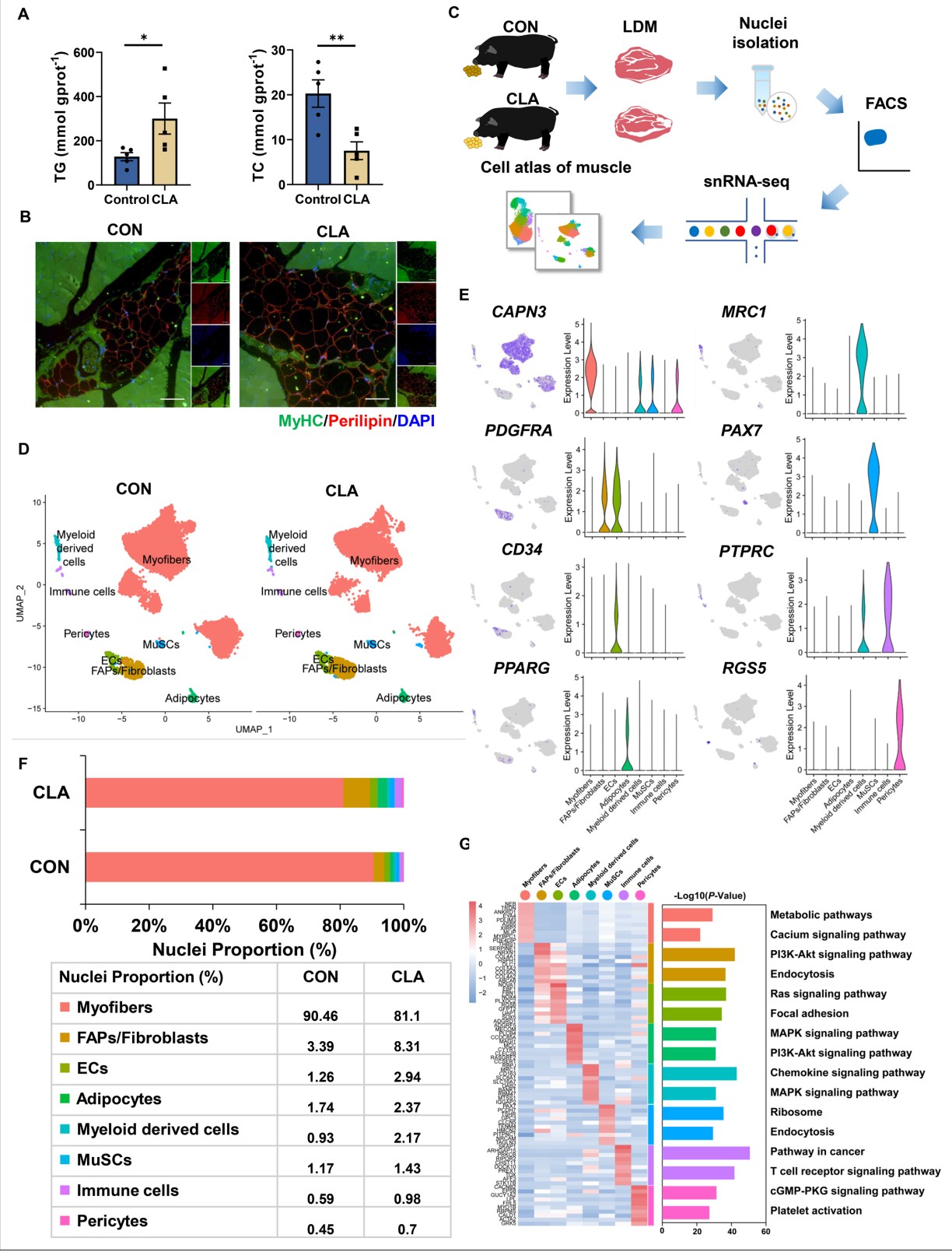

**Figure 1.** Single-nucleus RNA-sequencing (snRNA-seq) identifies distinct cell populations after conjugated linoleic acids (CLAs) treatment in pig muscles. (**A**) TG and TC content of *longissimus dorsi muscle* (LDM) tissues in control and CLAs groups (n=5). (**B**) LDM tissues stained with the adipogenic marker perilipin (red), muscle fiber marker MyHC (green), and DAPI (blue) in different groups. Scale bars, 200 and 100 μm, respectively. (**C**) Scheme of the experimental design for snRNA-seq on different muscles. (**D**) Uniform Manifold Approximation and Projection (UMAP) visualization of all of the

*Figure 1 continued on next page*

*Figure 1 continued*

isolated single nuclei from Heigai pig muscles colored by cluster identity. (**E**) UMAP and violin plot displaying the expression of selected marker genes for each cluster in pigs. (**F**) Nuclear proportion in each cluster in pig muscles of control and CLAs groups. Each cluster is color-coded. (**G**) Left, heatmap showing the top 10 most differentially expressed genes between cell types identified. Right, KEGG enrichment for marker genes of each cell type in muscles. Each lane represents a subcluster. Error bars represent SEM. *p<0.05, **p<0.01, two-tailed Student's t-test.

The online version of this article includes the following figure supplement(s) for figure 1:

**Figure supplement 1.** Conjugated linoleic acid (CLA) upregulated the expression of adipogenic-related genes in muscles.

## Clustering and RNA velocity analysis revealed subpopulations and cellular dynamics of adipocytes after CLAs treatment

To investigate the regulatory mechanism of CLAs in IMF deposition, we next performed a subset analysis on adipocytes. Our previous study had discovered there are three subclusters in adipocytes nuclei of Laiwu pigs (*Wang et al., 2023b*). In this study, we also characterized three subpopulations including SCD$^+$/DGAT2$^+$ adipocytes, FABP5$^+$/SIAH1$^+$ adipocytes, and PDE4D$^+$/PDE7B$^+$ adipocytes according to the most DEGs (*Figure 3A*, *Figure 3—figure supplement 1A*). Interestingly, we discovered CLAs increased the amounts of SCD$^+$/DGAT2$^+$ adipocytes (*Figure 3B*) and the proportion of SCD$^+$/DGAT2$^+$ adipocytes (79.37% vs 82.31%) but decreased the proportion of PDE4D$^+$/PDE7B$^+$ adipocytes (14.81% vs 11.19%) (*Figure 3—figure supplement 1B*). We also found CLAs increased the expression of SCD$^+$/DGAT2$^+$ adipocytes marker genes (*SCD* and *ARHGAP31*) and FABP5$^+$/SIAH1$^+$ adipocytes marker genes (*SIAH1* and *COX1*) but decreased the expression of PDE4D$^+$/PDE7B$^+$ adipocytes marker genes (*PDE4D*) in adipocytes (*Figure 3C*). Similarly, we also discovered *SCD* and *DGAT2* expression was remarkably increased in LDM after feeding with CLAs (*Figure 3D*). Meanwhile, immunofluorescence results showed more SCD1$^+$ adipocytes in the LDM of the CLAs group (*Figure 3E*). The top 10 most DEGs were displayed in the heatmap and the bar plot showed the top 3 KEGG enrichment of DEGs between the three subclusters in LDM (*Figure 3F*). To explore the effects of CLAs on differentiated trajectory of adipocytes, we carried out the RNA velocity analysis of adipocytes (*Figure 3—figure supplement 1C*). RNA velocity results showed the differentiated trajectory of mature adipocytes in muscles of Heigai pigs (*Figure 3G*). Transcriptional dynamics of *PDE4D* and *CAPN3* were shown in *Figure 3—figure supplement 1D* based on RNA velocity analysis. These results indicated in mature adipocytes, PDE4D$^+$/PDE7B$^+$ adipocytes and FABP5$^+$/SIAH1$^+$ adipocytes could differentiate into SCD$^+$/DGAT2$^+$ adipocytes (*Figure 3H*).

## The verification of differentiated trajectories of adipocytes by using Laiwu pig models

To further verify the differentiated trajectory of adipocytes, we next carried out a pseudotemporal trajectory analysis and RNA velocity analysis of adipocytes by using our previous high IMF content pig models based on Monocle 2 and scVelo (*Wang et al., 2023b*; *Figure 4A*). In adipocytes of Laiwu pigs, high IMF content Laiwu pigs (HLW) group had the higher proportion of SCD$^+$/DGAT2$^+$ adipocytes but the lower proportion of PDE4D$^+$/PDE7B$^+$ adipocytes (*Figure 4B*). Meanwhile, immunofluorescence staining results also showed the more SCD1$^+$ adipocytes in HLW group (*Figure 4C*). The pseudotemporal trajectory and RNA velocity analysis verified that PDE4D$^+$/PDE7B$^+$ subclusters could differentiate into two different directions, SCD$^+$/DGAT2$^+$ and FABP5$^+$/SIAH1$^+$ subclusters with two bifurcations (*Figure 4D–F*). The pseudotemporal heatmap showed gene expression dynamics including *COX1*, *ANO4*, *PPARG*, *ADIPOQ*, *ACACA*, and *ELOVL6* at Point 2 (*Figure 4—figure supplement 1A*). UMAP plots also showed transcriptional dynamics of marker genes such as *FABP4*, *ADIPOQ*, *MYBPC1*, and *EYV4* (*Figure 4—figure supplement 1B–D*). Also, we discovered the expression of preadipocytes-related genes, including *PDGFRA*, *CD34*, *CD38*, and *WT1* was enriched in PDE4D$^+$/PDE7B$^+$ adipocytes, while the expression of mature adipocytes-related genes, including *FABP4*, *ADIPOQ*, *LIPE*, *PLIN1*, *PPARG*, and *AGT* was enriched in SCD$^+$/DGAT2$^+$ and FABP5$^+$/SIAH1$^+$ adipocytes (*Figure 4G*). Hence, the differentiated trajectory of mature adipocytes in muscles was displayed in *Figure 4H* based on above results, which showed that PDE4D$^+$/PDE7B$^+$ adipocytes could differentiate into SCD$^+$/DGAT2$^+$ and FABP5$^+$/SIAH1$^+$ adipocytes, and FABP5$^+$/SIAH1$^+$ adipocytes can also differentiate into SCD$^+$/DGAT2$^+$ adipocytes. Additionally, the expression of *SCD*, *DGAT2*, *FABP5*, and *SIAH1* was upregulated but *PDE4D* and *PDE7B* expression were downregulated in HLW group (*Figure 4I*). These

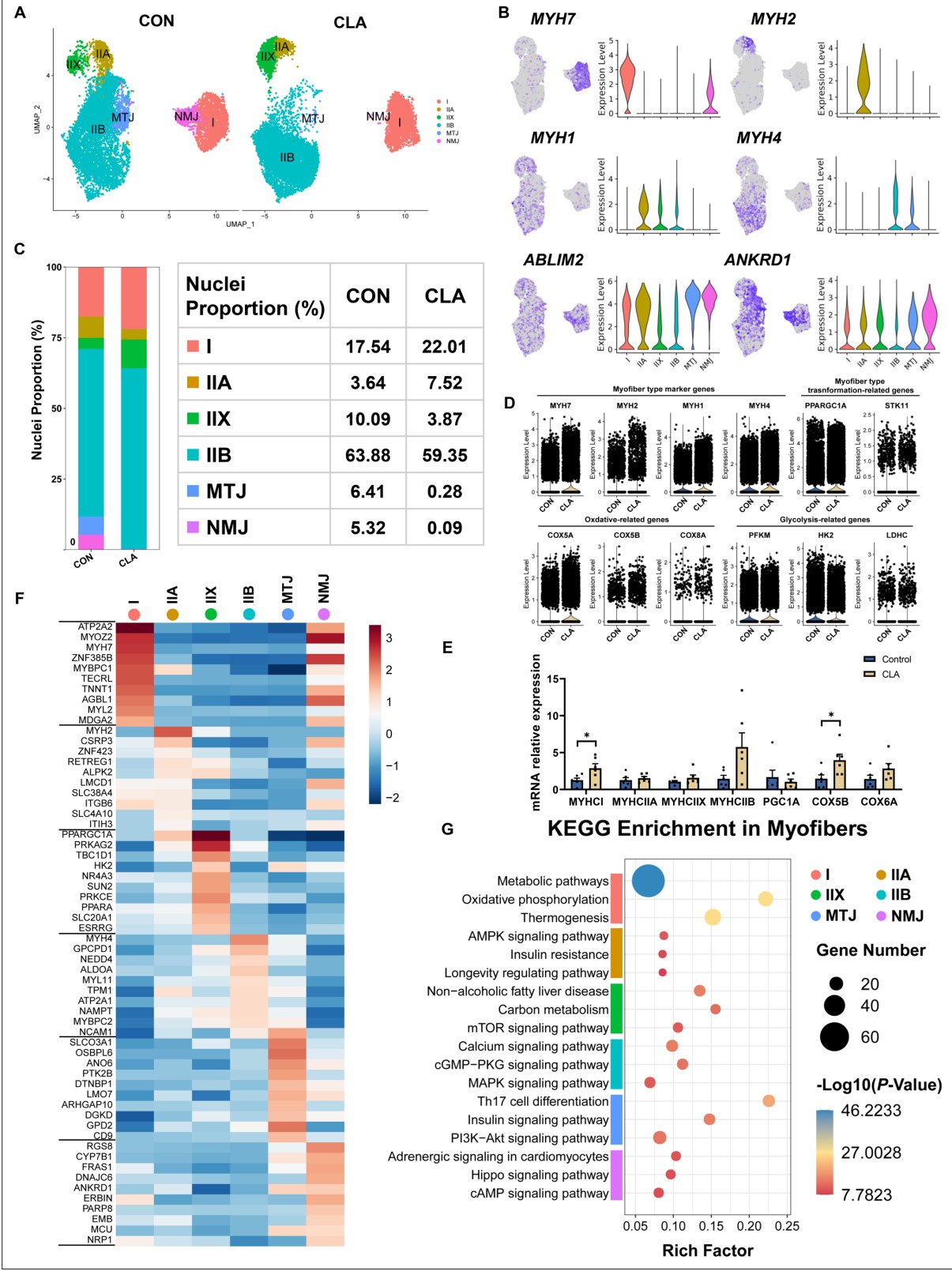

**Figure 2.** Cell and transcriptional heterogeneity in myofibers. (**A**) Uniform Manifold Approximation and Projection (UMAP) plot showing six subclusters of the isolated single nuclei from the control and conjugated linoleic acids (CLAs) muscles. (**B**) UMAP and violin plot displaying the expression of selected marker genes for each subcluster. (**C**) Cell proportion in each subcluster in different groups. Each cluster is color-coded. (**D**) Violin plot showing the expression of myofiber-type marker genes (*MYH7, MYH2, MYH1,* and *MYH4*), myofiber-type transformation-related genes (*PPARGC1A* and *STK11*),

*Figure 2 continued on next page*

*Figure 2 continued*

oxidation-related genes (*COX5A*, *COX5B*, and *COX8A*), and glycolysis-related genes (*PFKM*, *HK2*, and *LDHC*) after CLAs treatment. (**E**) The mRNA expression of myofiber-type-related genes in *longissimus dorsi muscle* (LDM) muscles after different treatment (n=6). (**F**) Heatmap representing the top 10 most differently expressed genes between cell subclusters identified. (**G**) KEGG enrichment for marker genes of each cell type in myofibers. I, type I myonuclei; IIa, type IIa myonuclei; IIx, type IIx myonuclei; IIb, type IIb myonuclei; MTJ, myotendinous junction nuclei; NMJ, neuromuscular junction nuclei. Error bars represent SEM. *p<0.05, two-tailed Student's t-test.

data indicated the differentiated trajectory of mature adipocytes in muscle of pigs and the percentage of SCD$^+$/DGAT2$^+$ and FABP5$^+$/SIAH1$^+$ subclusters was higher in HLW group.

## Transcriptional dynamics of glycerophospholipid metabolism in high IMF deposition pigs

Our previous studies have discovered the changes in glycerophospholipid metabolism in muscles after CLA treatment (*Wang et al., 2022b*) and in high IMF content Laiwu pigs (*Wang et al., 2023b*). To further investigate the transcriptional dynamics of glycerophospholipid metabolism in high IMF deposition pigs, we then compare the gene program in Heigai pigs and Laiwu pigs. After CLA treatment, the snRNA-seq dataset revealed slight differences in the expression of genes involved in glycerophospholipid metabolism across different groups and subclusters (*Figure 4—figure supplement 2A–C*). Also, in Laiwu pigs, there are differences in gene program involved in the glycerophospholipid metabolism between two groups (*Figure 4—figure supplement 2D–F*). Interestingly, we found *LCLAT1* was enriched in PDE4D$^+$/PDE7B$^+$ subcluster, and *AGPAT3* and *AGPAT5* was enriched in SCD$^+$/DGAT2$^+$ subcluster (*Figure 4-figure supplement 2B and Figure 4-figure supplement 2E*). We also discovered the increase of diglycerides and phosphatidylinositols and the decrease of phosphatidic acids and phosphatidylethanolamines in high IMF deposition pigs might be due to the changes in *AGPAT3*, *AGPAT4*, *AGPAT5*, *CEPT1*, and *CDIPT1* (*Figure 4-figure supplement 2C and Figure 4-figure supplement 2F*). These data revealed the significant differences in lipid composition and distribution in LDM of high IMF deposition pigs might be due to the different expression levels of glycerophospholipid metabolism-related genes.

## Cell-cell communication analysis showed the interaction between adipocytes and FAPs/fibroblasts

To further explore the association between adipocytes nuclei and other cell clusters, we applied cell-cell communication analysis by using CellPhoneDB on eight cell types in muscles of Heigai pigs and found adipocytes mainly interacted with ECs, FAPs/fibroblasts, MuSCs, and pericytes (*Figure 5A*). In addition, dot plot represented stronger communication from adipocytes to other subclusters through LRP6, FGFR1, and COL4A2 pathways in LDM of Heigai pigs (*Figure 5C*). Next, we also applied cell-cell communication analysis by using CellPhoneDB on nine cell types in muscles of Laiwu pigs and found adipocytes mainly interacted with SPs, FAPs/fibroblasts, ECs, and pericytes (*Figure 5B*). Also, dot plot showed stronger communication from adipocytes to other subclusters through COL6A3, LAMC1, and THBS1 pathways in LDM of Laiwu pigs (*Figure 5D*). These results suggested adipocytes have a tight association with FAPs/fibroblasts.

## Characterization of FAPs after CLAs treatment through clustering and pseudotime analysis

Previous studies have demonstrated that FAPs could differentiate into mature adipocytes and are the main cell sources of IMF cells (*Joe et al., 2010*; *Uezumi et al., 2010*). To further investigate the effects of CLAs on occurrence mechanism of IMF deposition, we then performed subcluster and pseudotemporal trajectory analysis on FAPs/fibroblasts. First, UMAP plots displayed the cell distribution in different subpopulations of FAPs/fibroblasts (*Figure 6A*) and we identified three subpopulations in FAPs/fibroblasts based on the expression of marker genes, including FAPs (*PDGFRA*), fibroblasts (*COL1A1*), and PDE4D$^+$/PDE7B$^+$ subclusters (*PDE4D* and *PDE7B*) (*Figure 6B*). In CLA group, we found the proportion of FAPs (70.40% vs 37.06%) and PDE4D$^+$/PDE7B$^+$ (4.11% vs 2.18%) was higher than that in CON group but the proportion in fibroblasts (25.49% vs 60. 76%) was lower (*Figure 6C*). The top 10 most DEGs between the three subpopulations were shown in the heatmap and the bar plot displayed the

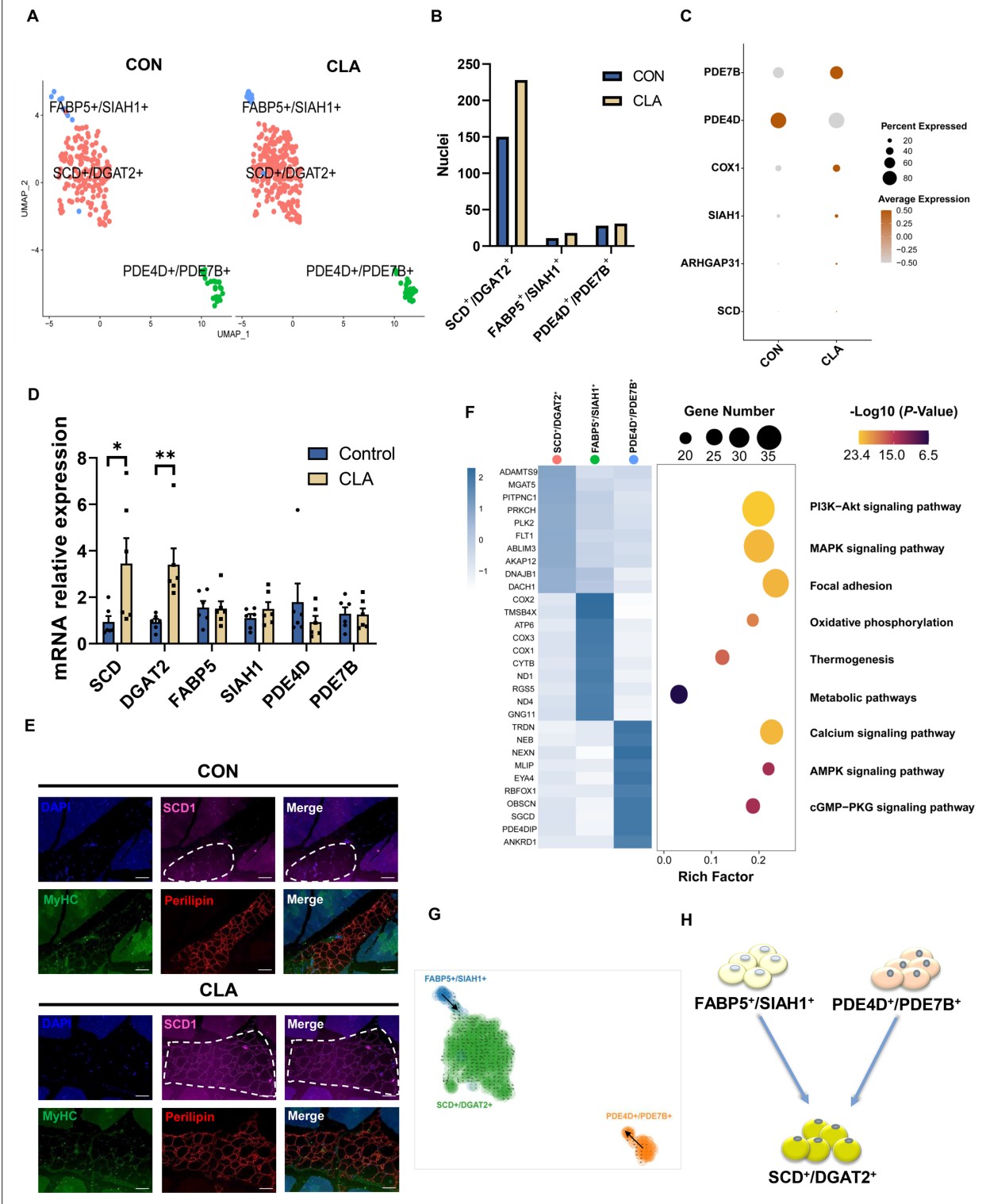

**Figure 3.** Clustering and transcriptional dynamics of adipocytes. (**A**) Uniform Manifold Approximation and Projection (UMAP) plot displaying the isolated single nuclei in three subclusters of adipocytes. (**B**) Bar plot displaying the cell amounts in each subcluster in different groups. (**C**) Dot plot showing the expression of three subcluster marker genes in muscle nuclei of Heigai pigs. (**D**) The mRNA expression of three subcluster marker genes in *longissimus dorsi muscle* (LDM) muscles after different treatment (n=6). (**E**) LDM tissues stained with the adipogenic marker perilipin (red), muscle fiber marker MyHC (green), SCD1 (pink), and DAPI (blue) in different groups. Scale bars, 100 μm. (**F**) Left, heatmap showing the top 10 most differentially expressed genes between cell types identified. Right, KEGG enrichment for marker genes of each cell type in muscles. (**G**) Unsupervised pseudotime trajectory of the three subtypes of adipocytes by RNA velocity analysis. Trajectory is colored by cell subtypes. The arrow indicates the direction of cell

*Figure 3 continued on next page*

Figure 3 continued

pseudotemporal differentiation. (**H**) Scheme of the differentiation trajectories in mature adipocytes. Error bars represent SEM. *p<0.05, **p<0.01, two-tailed Student's t-test.

The online version of this article includes the following figure supplement(s) for figure 3:

**Figure supplement 1.** Clustering analysis of adipocytes nuclei.

significant enrichment of the signaling pathways in muscles by using KEGG enrichment analyses, and we also found calcium and cGMP-PKG signaling pathways were enriched in PDE4D$^+$/PDE7B$^+$ subclusters (*Figure 6D*). Besides, dot plot showed CLAs upregulated the expression of preadipocytes-related genes including *CD38* and *CD34*, adipogenic master genes including *ADIPOQ*, *FABP4*, *PLIN1*, and *LIPE*, mature adipocyte marker genes including *CEBPA*, and lipid metabolism-related genes including *LPL*, *ELOVL4*, *ACAA2*, and *HACD2* in FAPs/fibroblasts (*Figure 6—figure supplement 1A*). To further explore FAPs' differentiated trajectory, we applied a pseudotemporal trajectory analysis and RNA velocity analysis of FAPs/fibroblasts. According to the results, we found FAPs could differentiate into PDE4D$^+$/PDE7B$^+$ and fibroblasts subpopulations (*Figure 6E*, *Figure 6—figure supplement 1C*). The pseudotemporal heatmap also displayed transcriptional dynamics including *TIMP3* and *THBS1* at Point 1 (*Figure 6F*). UMAP plots also showed transcriptional dynamics of marker genes in three subpopulations such as *ITGA5*, *HSPH1*, and *ETF1* (*Figure 6—figure supplement 1B–E*). To further identify the differentiated trajectory of IMF cells, we isolated primary FAPs from pigs and found the expression of *FABP4*, *ADIPOQ*, *SCD*, and *DGAT2* was significantly increased but *PDE4D* expression was significantly downregulated during adipogenic differentiation in vitro (*Figure 6G*). Besides, the expression of *FABP5* and *SIAH1* was first significantly increased then significantly decreased during adipogenic differentiation (*Figure 6G*). Hence, FAPs may first differentiate into PDE4D$^+$/PDE7B$^+$ preadipocytes and then differentiate into PDE4D$^+$/PDE7B$^+$ adipocytes (*Figure 6H*). These results displayed the differentiated trajectory of preadipocytes into mature adipocytes in pig muscles, suggesting that CLA might influence this process and subsequently affect IMF deposition.

## CLAs promoted FAPs' directed differentiation into SCD$^+$/DGAT2$^+$ subclusters

To further explore the cytological mechanism of CLAs regulating IMF deposition, we next investigate the regulatory effects of CLAs on the differentiation trajectory of FAPs differentiated into adipocytes. First, to explore the role of CLAs in the adipogenic differentiation of FAPs, we isolated primary FAPs from piglets and induced these adipogenic differentiation. Nile Red staining and OD490 results revealed CLAs can promote the adipogenic differentiation of FAPs after adipogenic differentiation for 8 days in vitro (*Figure 7—figure supplement 1A–B*). Besides, the mRNA expression of *SCD*, *SIAH1*, and adipogenic genes, including *FABP4*, *PPARG*, and *FASN*, were significantly upregulated but that of *PDE4D* was significantly downregulated (*Figure 7C*). The protein levels of FABP4 and SCD1 were significantly upregulated and that of PDE4D were significantly downregulated (*Figure 7D*). In addition, the marker genes' expression at key points of adipogenic differentiation such as *ADIPOQ*, *ELOVL6*, *ACACA*, *ARBB1*, *NEB*, and *MYBPC1* were significantly upregulated and that of *THBS1* and *TIMP3* were significantly downregulated (*Figure 6—figure supplement 1A*). Interestingly, we next found MAPK signaling pathways were enriched in adipocytes nuclei, especially SCD$^+$/DGAT2$^+$ subcluster (*Figure 7E*). The expression of MAPK signaling pathway including ERK, c-Jun N-terminal kinase (JNK), and p38 signaling pathway-related genes were changed in muscle nuclei (*Figure 7F*, *Figure 6—figure supplement 1B and C*). Furthermore, we observed that the protein levels of JNK phosphorylation were significantly decreased after CLA treatment during FAPs' adipogenic differentiation (*Figure 7G*). Hence, we next used JNK activator anisomycin to explore the regulatory mechanism (*Figure 7H*). Oil Red O staining result showed that after 24 hr 20 nM anisomycin treatment, the increased adipogenic differentiation of FAPs by CLA were significantly inhibited (*Figure 7I*). Also, CLA+anisomycin group had the lower OD 490 value compared with the CLA group (*Figure 7J*). Nile Red staining result also showed that anisomycin significantly inhibited the enhanced lipid droplets in FAPs after CLA treatment (*Figure 7K*). MAP2K4 expression significantly upregulated and the expression of *FABP4* and *SCD* significantly decreased after anisomycin treatment (*Figure 7L*). These data indicated that CLAs

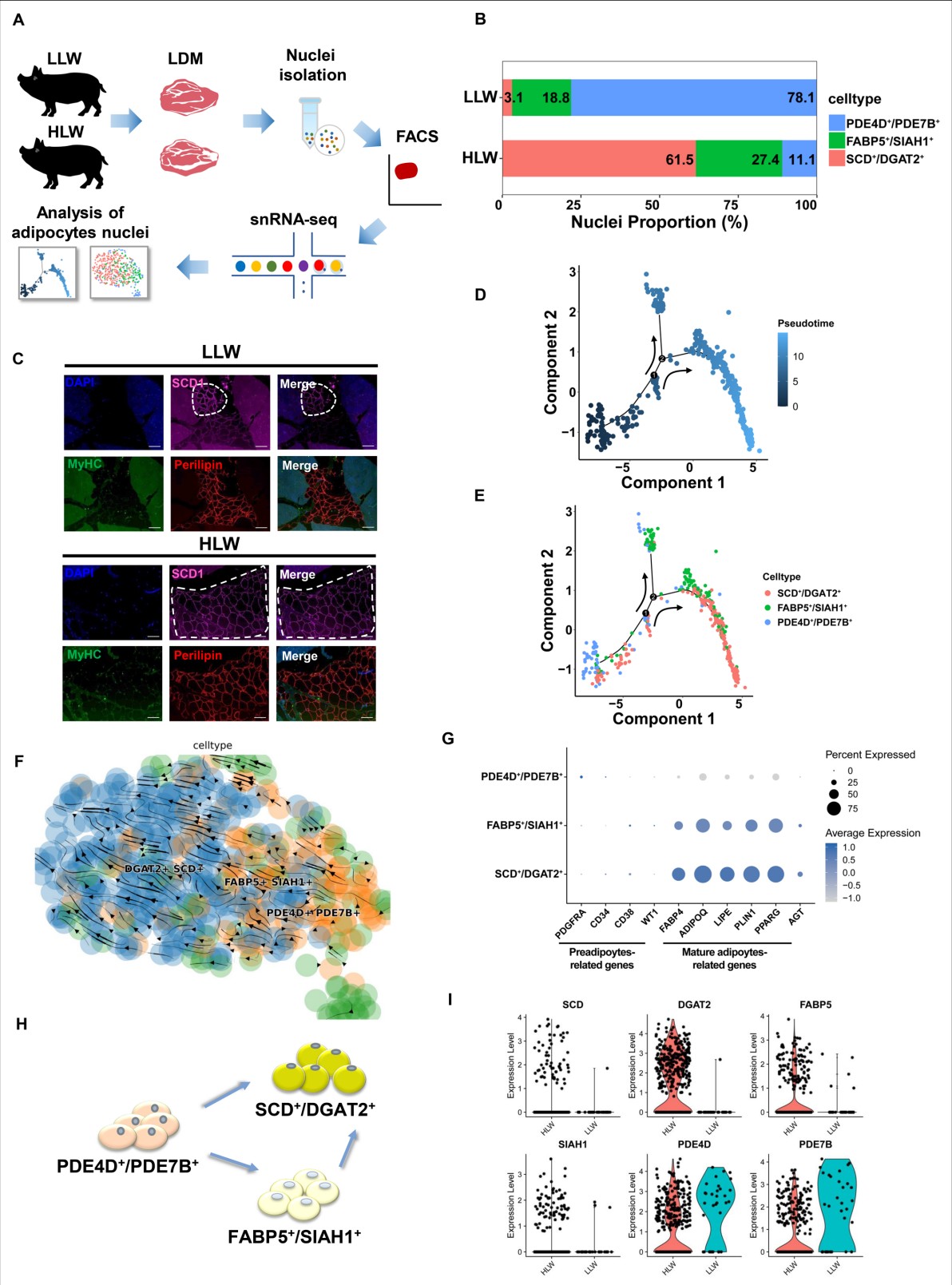

**Figure 4.** Pseudotemporal and differentiated trajectories of adipocytes in high intramuscular fat (IMF) content Laiwu pig muscles. (**A**) Scheme of the experimental design for single-nucleus RNA-sequencing (snRNA-seq) on adipocytes of high IMF content Laiwu pig muscles. (**B**) Cell proportion of adipocytes subclusters in high IMF content Laiwu pigs (HLW) and low IMF content Laiwu pigs (LLW) groups. Each cluster is color-coded. (**C**) *Longissimus dorsi muscle* (LDM) tissues stained with the adipogenic marker perilipin (red), muscle fiber marker MyHC (green), SCD1 (pink), and DAPI (blue) in HLW

*Figure 4 continued on next page*

*Figure 4 continued*

and LLW groups. Scale bars, 100 µm. (**D–E**) Pseudotime ordering of all of the adipocytes of subclusters DGAT2$^+$/SCD$^+$, FABP5$^+$/SIAH1$^+$, and PDE4D$^+$/PDE7B$^+$. Each dot represents one nucleus (color-coded by its identity), and each branch represents one cell state. Pseudotime is shown colored in a gradient from dark to light blue, and the start of pseudotime is indicated. Activation of the PDE4D$^+$/PDE7B$^+$ cluster can lead to DGAT2$^+$/SCD$^+$ and FABP5$^+$/SIAH1$^+$ fate. (**F**) Unsupervised pseudotime trajectory of the three subtypes of adipocytes by RNA velocity analysis. Trajectory is colored by cell subtypes. The arrow indicates the direction of cell pseudotemporal differentiation. (**G**) Dot plot showing the expression of preadipocytes and mature adipocytes-related genes in different subclusters. (**H**) Scheme of the differentiation trajectories in mature adipocytes of Laiwu pigs. (**I**) Violin plot showing the expression of three subcluster marker genes in different groups.

The online version of this article includes the following figure supplement(s) for figure 4:

**Figure supplement 1.** Pseudotime trajectory analysis of adipocytes nuclei by RNA velocity in high IMF content Laiwu pigs (HLW) pigs.

**Figure supplement 2.** Comparison of gene programs involved in glycerophospholipid metabolism.

may promote the directed differentiation of FAPs into SCD$^+$/DGAT2$^+$ subclusters via inhibiting JNK signaling pathway.

## Discussion

CLAs can serve as a nutritional intervention to regulate lipid deposition in skeletal muscle of human according to clinic trials (*van Vliet et al., 2020*). In animal production, CLAs could regulate meat quality in pigs and cattle, especially improve IMF content (*Wang et al., 2022b*; *Zhang et al., 2016*). These studies pointed out that the CLAs play a vital role in regulating fat infiltration in skeletal muscle. However, to date, the cellular mechanism of CLAs that regulates lipid deposition has not been studied. Here, we utilized the 10x Genomics platform to identify the cell heterogeneity and transcriptional changes in muscles after CLAs treatment based on pig models. This study revealed the effects of CLAs on cell populations and molecular characteristics of muscles and highlighted the cytological mechanism of CLAs that regulates pork quality in skeletal muscle.

CLAs are commonly found in ruminant animals and dairy products; they are a class of positional and geometric isomers of linoleic acid with conjugated double bonds. CLAs not only have anticancer, antihypertension, anti-adipogenic, and antidiabetic effects, but also can improve muscle function and decrease body fat percentage. For example, adding 3.2 g/day CLAs significantly increased muscle mass in higher body fat percentage Chinese adults (*Chang et al., 2020*). After 0.9 g/day CLAs supplementation, body weight variation and muscle mass significantly increased and body fat percentage variation decreased in student athletes (*Terasawa et al., 2017*). LC-MS metabolomics results discovered CLAs changed 57 metabolites which enriched in lipids/lipid-like molecules in plasma of humans (*He et al., 2022*). However, another study found that for sedentary older adults, CLAs had no significant influence on muscle anabolic effects (*van Vliet et al., 2020*). Therefore, the specific influences of CLAs on skeletal muscle are still disputed and the function on lipid deposition in human skeletal muscle needs further investigation. In animal models, many studies have demonstrated the important effects of CLAs on regulating fat accumulation in skeletal muscles. In porcine models, our foregoing studies have discovered that adding CLAs into the pig diet could significantly increase IMF contents in LDM of lean pig breeds and Heigai pigs (*Wang et al., 2021*; *Wang et al., 2022b*). *Zhang et al., 2016*, found 2% dietary CLAs significantly increased IMF deposition and reduced subcutaneous fat deposition in cattle. However, in mouse model, 0.5% mixed isomer CLAs did not lead to lipid accumulation in muscle of mice (*Kanosky et al., 2013*). Hence, CLAs supplementation positively affect IMF deposition in muscles of pigs and ruminants but the effects of CLAs may have species-specific. In our study, we identified eight cell types in skeletal muscles of Heigai pigs, including myofibers, FAPs/fibroblasts, ECs, adipocytes, immune cells, MuSCs, myeloid-derived cells, and pericytes. Recently, a variety of single cell studies have been performed on mouse/human skeletal muscles and they also identified some cell populations such as myofibers, FAPs/fibroblasts, ECs, adipocytes, immune cells, MuSCs, myeloid-derived cells, tenocytes, SPs, and pericytes (*Petrany et al., 2020*; *Xu et al., 2023*; *Xu et al., 2021*). We also found CLAs improved TG content and increased the percentage of adipocytes in LDM. Previous study demonstrated there are three subclusters in adipocytes and the formation and deposition of IMF mainly relied on DGAT2$^+$/SCD$^+$ adipocytes and FABP5$^+$/SIAH1$^+$ adipocytes (*Wang et al., 2023b*). Specially, we found CLAs enhanced the percentage of SCD$^+$/DGAT2$^+$ subclusters. These indicated CLAs might improve IMF deposition through increasing SCD$^+$/DGAT2$^+$subpopulations of

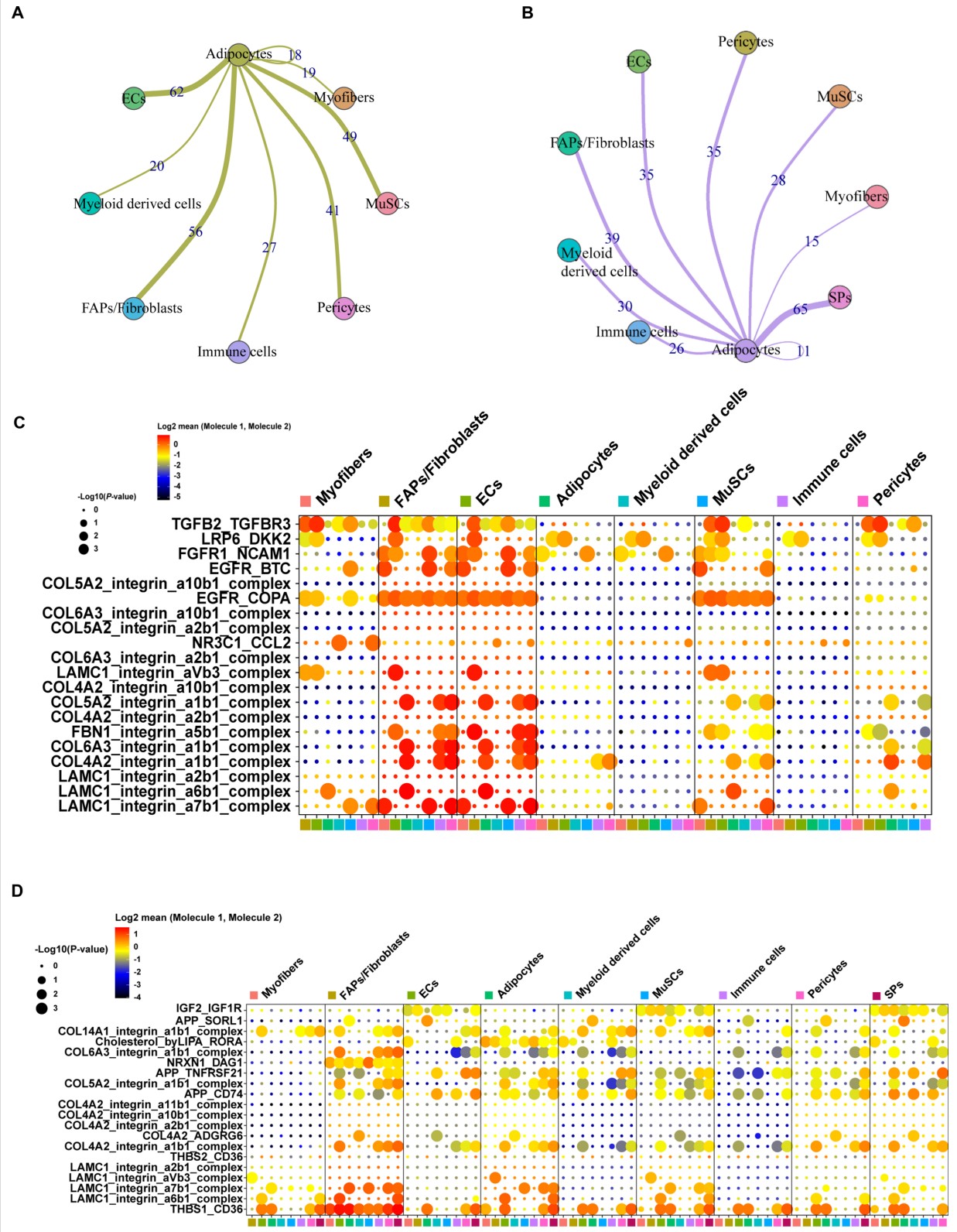

**Figure 5.** Cell-cell communication analysis of adipocytes in pig muscles. (**A**) Cell-cell communication analysis showed the network between adipocytes and other clusters in muscles of Heigai pigs. (**B**) Cell-cell communication analysis showed the network between adipocytes and other clusters in muscles of Laiwu pigs. (**C**) Dot plot representing the gene expression and significance of the receptor-ligand relationship in different cell population in muscles of Heigai pigs. (**D**) Dot plot representing the gene expression and significance of the receptor-ligand relationship in different cell population in muscles of Laiwu pigs. The larger the circle, the smaller the p-value of the relationship in the corresponding cell population, the more significant it is.

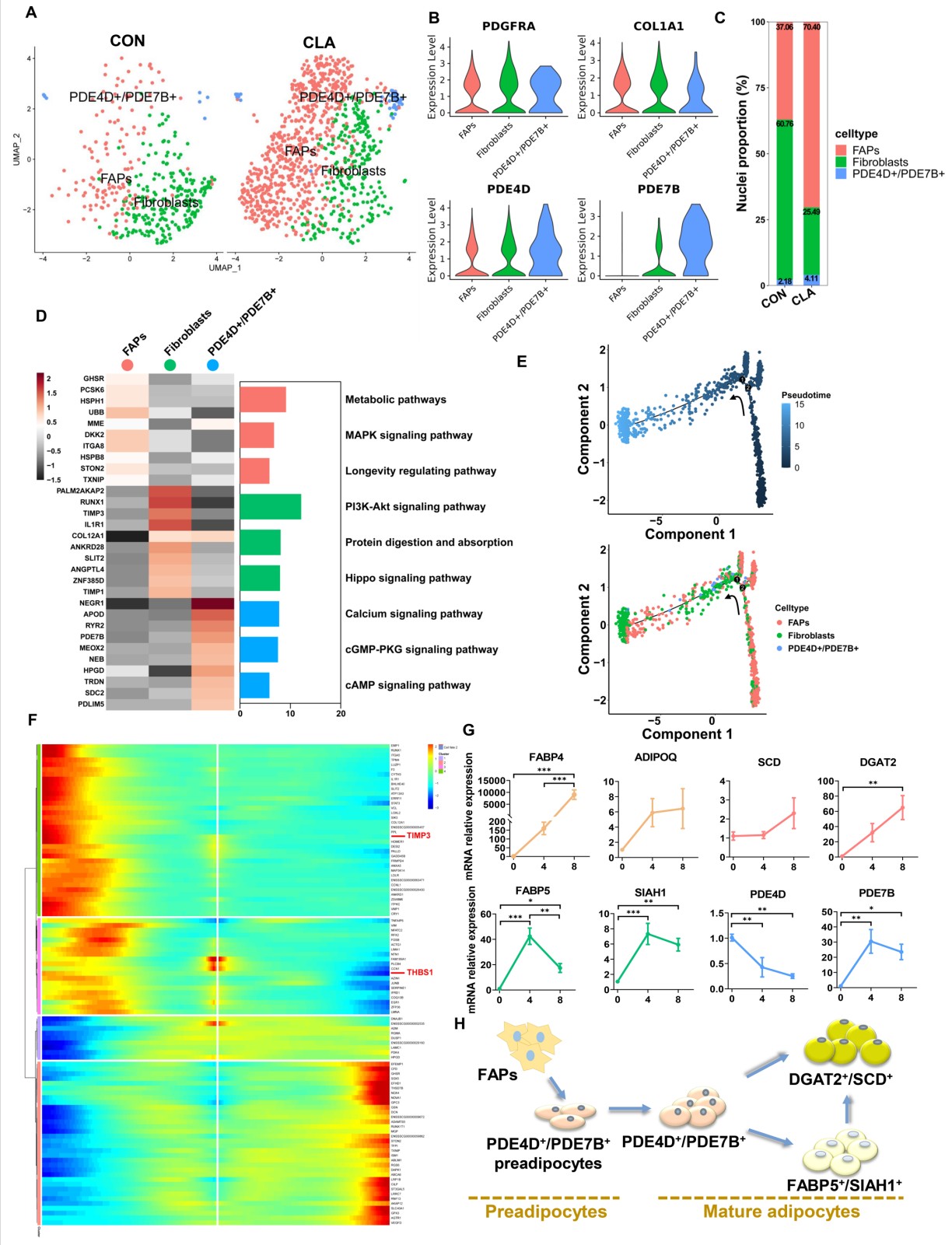

**Figure 6.** Clustering and pseudotemporal trajectories of fibro/adipogenic progenitors (FAPs). (**A**) Uniform Manifold Approximation and Projection (UMAP) plot showing three subclusters of the isolated single nuclei from control and conjugated linoleic acids (CLAs) muscle. (**B**) Violin plot displaying the expression of selected marker genes for each subcluster. (**C**) Cell proportion in each subcluster in different groups. Each cluster is color-coded. (**D**) Left, heatmap showing the top 10 most differentially expressed genes between cell types identified. Right, KEGG enrichment for marker genes of each

*Figure 6 continued on next page*

*Figure 6 continued*

cell type in muscles. (**E**) Pseudotime ordering of all of the FAP/fibroblast of subcluster FAPs, fibroblasts, and PDE4D⁺/PDE7B⁺. Each dot represents one nucleus (color-coded by its identity), and each branch represents one cell state. Pseudotime is shown colored in a gradient from dark to light blue, and the start of pseudotime is indicated. Activation of the FAP cluster can lead to fibroblast fate or PDE4D⁺/PDE7B⁺ fate. (**F**) Pseudotemporal heatmap showing gene expression dynamics for significant marker genes. Genes (rows) were clustered into three modules, and cells (columns) were ordered according to pseudotime in different groups. (**G**) The expression of adipogenesis and three subcluster marker genes in differentiated FAPs in different differentiation stage (n=6). (**H**) Scheme of the differentiation trajectories of preadipocytes into mature adipocytes. Error bars represent SEM. *p<0.05, **p<0.01, ***p<0.001, two-tailed Student's t-test.

The online version of this article includes the following figure supplement(s) for figure 6:

**Figure supplement 1.** Pseudotime trajectory analysis of fibro/adipogenic progenitors (FAPs) nuclei by RNA velocity.

adipocytes. Our findings could provide a foundation for using nutritional strategies to increase pork quality, especially IMF deposition. However, the specific function of CLAs on fat infiltration and deposition of skeletal muscle in people and rodents needs further study.

Skeletal muscle contains slow and fast muscle fibers and there are four major muscle fiber types in mice, including slow muscle fibers with type I muscle fibers (*Myh7*), fast muscle fibers with type IIA muscle fibers (*Myh2*), type IIX muscle fibers (*Myh1*), and type IIB muscle fibers (*Myh4*) (***Dos Santos et al., 2020***; ***Petrany et al., 2020***). In pigs, we illustrated three myofiber composition, including type I, type IIA, and type IIB in skeletal muscles by ST technology (***Jin et al., 2021***). In this study, we identified six different subpopulations in myofibers, including I myofibers, IIA myofibers, IIX myofibers, IIB myofibers, MTJ, and NMJ. IIB myofibers had the highest percentage in pig muscles. MTJ are known to exhibit structural specialization; NMJ are responsible for formation and maintenance of the synaptic apparatus, and previous studies have identified these cell populations in murine skeletal muscles by using snRNA-seq (***Dos Santos et al., 2020***; ***Petrany et al., 2020***). We also identified MTJ and NMJ cell populations in LDM of Laiwu pigs (***Wang et al., 2023b***). These results indicate that MTJ and NMJ cell populations also exist in porcine skeletal muscles. In our previous study, we also found the percentage of type IIa myofibers had an increased tendency and type IIb myofibers had a decreased tendency in high IMF content Laiwu pigs (***Wang et al., 2023b***). These results suggest that IMF content is closely related to muscle fiber type. Besides, slow muscle fibers, always called slow-twitch oxidative muscle fibers like I myofibers, have higher activities in mitochondrial oxidative metabolic enzymes and myoglobin while fast muscle fibers, always called fast-twitch glycolytic muscle fibers like II myofibers, have higher levels of glycolytic enzymes and glycogen (***Schiaffino and Reggiani, 2011***). Similarly, we also found I myofibers enriched in metabolic pathways, oxidative phosphorylation, and thermogenesis. In recent years, studies have focused on exploring the influences of CLAs on regulating muscle fiber type. In commercial pigs, the MyHC I mRNA abundance were improved in LDM of the CLAs group (***Men et al., 2013***). In mice, t10, c12-CLAs, but not c9, t11-CLAs can increase oxidative skeletal muscle fiber type in gastrocnemius muscle and C2C12 myoblasts (***Duan et al., 2021***). CLAs have been found to prevent sarcopenia by maintaining redox balance during aging, actively regulating mitochondrial adaptation, improving muscle metabolism, and inducing hypertrophy of type IIX myofibers after endurance exercise (***Barone et al., 2017***; ***Chen et al., 2018***). Also, we discovered CLAs enhanced the percentage of I and IIA myofibers but reduced the percentage of IIB myofibers. Previous study has found PPARγ coactivator-1α (PGC1α) serves a valuable role in skeletal muscle metabolism and is a master regulator of oxidative phosphorylation genes and could regulate muscle fiber-type transformation (***Handschin and Spiegelman, 2011***). In our study, the PGC1α expression was also increased after CLA treatment in myofiber. These results suggested CLAs can promote glycolytic skeletal muscle fiber types switching into oxidative skeletal muscle fiber types through upregulating PGC1α expression.

Numerous studies have discovered the cell sources of IMF cells and found several cell subsets lead to ectopic IMF formation and deposition, including SCs, Myf5⁺ MSCs, FAPs, ECs, pericytes, fibroblasts, myeloid-derived cells, SPs, and PICs (***Sciorati et al., 2015***; ***Xu et al., 2021***). In this study, we found CLAs increased the percentage of preadipocytes such as FAPs, ECs, myeloid-derived cells, and pericytes. Importantly, FAPs are the major source of IMF cells (***Joe et al., 2010***; ***Uezumi et al., 2010***) and our previous study also verified the adipogenic capacity of FAPs in 2D and 3D culture models (***Wang et al., 2023b***). Xu et al. found that FAPs serve as a cellular interaction hub in skeletal muscle of pigs (***Xu et al., 2023***). We used pseudotemporal trajectory and RNA velocity analysis combined with in vitro study to investigate that FAPs could first differentiate into PDE4D⁺/PDE7B⁺ adipocytes and then

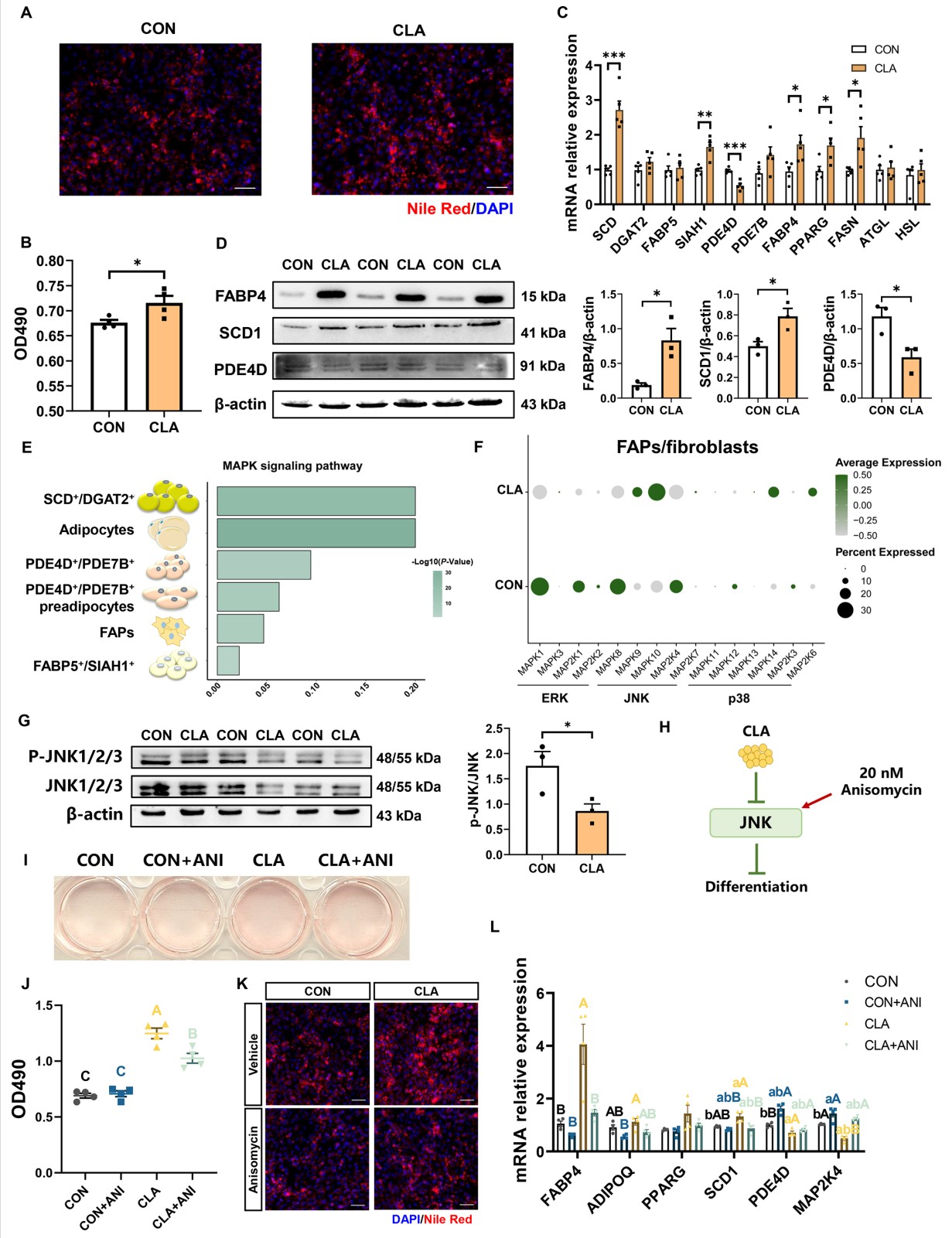

**Figure 7.** The cytological mechanism of conjugated linoleic acids (CLAs) regulates fibro/adipogenic progenitors (FAPs) differentiation. (**A**) Differentiated FAPs stained with Nile Red (red) and DAPI (blue) in different groups. Scale bars, 200 and 100 μm, respectively. (**B**) OD490 levels of total lipids in differentiated FAPs after different treatment (n=4). (**C**) The mRNA expression of three subcluster marker genes and adipogenic marker genes in differentiated FAPs after different treatment (n=5). (**D**) Protein levels of FABP4, SCD1, and PDE4D were detected by western blot. (**E**) MAPK signaling

*Figure 7 continued on next page*

*Figure 7 continued*

pathway enrichment in different cells. (**F**) Dot plot showing the expression of MAPK signaling pathway-related genes after CLA treatment in FAPs/fibroblasts. (**G**) Protein levels of P-JNK and JNK were detected by western blot. (**H**) Scheme of CLAs regulating the differentiation trajectories of FAPs into mature adipocytes. (**I**) Differentiated FAPs stained with Oil Red O in different groups after treating with 20 nM anisomycin. (**J**) OD490 levels of total lipids in differentiated FAPs after different treatment (n=4). (**K**) Differentiated FAPs stained with Nile Red (red) and DAPI (blue) after 20 nM anisomycin treatment. Scale bars, 200 µm. (**L**) The mRNA expression of adipogenic-related genes in differentiated FAPs after different treatment (n=4). Error bars represent SEM. *p<0.05, **p<0.01, ***p<0.001, two-tailed Student's t-test and one-way ANOVA. Letters represent statistical significance. Lowercase letters represent p<0.05 and uppercase letters represent p<0.01.

The online version of this article includes the following source data and figure supplement(s) for figure 7:

**Source data 1.** PDF file containing original western blots for *Figure 7D and G*.

**Source data 2.** Original files for western blot analysis displayed in *Figure 7D and G*.

**Figure supplement 1.** Changes in MAPK signaling pathway in muscle nuclei.

differentiate into DGAT2+/ SCD+ and FABP5+/SIAH1+ adipocytes. However, the regulatory mechanism of FAPs' directional differentiation still needs to be further explored. In vitro studies demonstrated trans-10, cis-12 CLAs inhibited skeletal muscle differentiation in C2C12 cells and inhibited 3T3-L1 adipocyte adipogenesis (*Hommelberg et al., 2010*; *Yeganeh et al., 2016*). However, the influences of CLAs on regulating the adipogenic differentiation of FAPs remain unclear. In this study, we found CLAs facilitated FAPs differentiating into SCD+/DGAT2+ adipocytes. Previous studies have found mice orally treated with CLAs mixture upregulated *Scd1* expression in muscle (*Parra et al., 2012*). Besides, SCD1 expression is modulated by mTOR signaling pathway in cancer cells (*Yi et al., 2020*; *Zhao et al., 2021*). Moreover, MAPK signaling pathways were enriched in adipogenic differentiation of FAPs after CLAs treatment. Previous studies have discovered JNK had negative effects on regulating the adipogenic differentiation of human MSCs (*Jang et al., 2015*) and FAPs could prevent skeletal muscle regeneration after muscle injury by ST2/JNK signaling pathways (*Yamakawa et al., 2023*). In this study, based on JNK signing pathway activator treatment and in vitro experiment, we found CLAs may promote FAPs' directed differentiation into SCD+/DGAT2+ adipocytes via inhibiting JNK signaling pathway. These results may provide new targets for treating human fat infiltrated diseases by nutritional strategies. However, we did not further explore the mechanistic action, and the downstream transcriptional regulators need to be discussed.

In a word, we provide detailed insights into the cytological mechanism of CLAs regulates fat infiltration in skeletal muscles based on pig models via using snRNA-seq. We analyzed the effects of CLAs on the cell heterogeneity and transcriptional dynamics in pig muscles and discovered CLAs could promote glycolytic muscle fiber types switching into oxidative muscle fiber types through regulating PGC1α. We also identified the differentiation trajectories of adipocytes and FAPs. Our data also demonstrated CLAs could promote FAPs differentiate into DGAT2+/SCD+ adipocytes via inhibiting JNK signaling pathway. This study provides a new way of developing nutritional strategies to combat myosteatosis and other muscle-related diseases and also offers potential opportunities to promote the utilization of pigs as animal models to study human diseases.

## Materials and methods
### Animals and samples

The Zhejiang University's Animal Care and Use Committee approved all procedures and housing (ZJU20170466). 56 Heigai pigs (average body weight: 85.58±10.39kg) were divided randomly into CON group (added 1% soybean oil) and CLA group (added 1% CLAs) for 40days (5days pre-feeding period and 35days formal test period). The nutritional levels and the feeding processes reported as we discussed previously (*Wang et al., 2022b*; *Wang et al., 2023a*). At the end of experiment, we collected LDM from the right side of the carcass for subsequent immunofluorescence staining, biochemical assay, and snRNA-seq analyses. For Laiwu pigs, based on the determination of IMF content in Laiwu pigs, we divided them into two groups: HLW group and low IMF content Laiwu pigs group. The two most representative samples from each group were selected for later snRNA-seq, and datasets generated from muscle of high IMF content pig samples were downloaded from the Genome Sequence Archive (*Chen et al., 2021*) in National Genomics Data Center (*CNCB-NGDC Members*

*and Partners, 2024*), China National Center for Bioinformation/Beijing Institute of Genomics, Chinese Academy of Sciences (GSA: CRA011059; https://ngdc.cncb.ac.cn/gsa) as we previously discussed (*Wang et al., 2023b*).

## TG and TC determination

The contents of TG and TC in LDM were determined by commercial kits (TG, E1025-105; TC, E1015-50) bought from Beijing APPLYGEN Gene Technology Co., Ltd.

## Immunofluorescence staining

The paraffin section was dewaxed and immersed in preheated sodium citrate, then placed in a microwave oven and heated for 15 min to perform antigen retrieval. The sections were fixed in 4% paraformaldehyde for 10 minutes at room temperature, permeabilized with 0.5% Triton X-100 for 10 minutes, and then blocked with blocking buffer (5% goat serum and 2% BSA) for 1 hour. Sections were then incubated overnight at 4°C with Perilipin (Abcam, ab16667, 1:500), MF20 (Developmental Studies Hybridoma Bank, 1:50), and SCD1 (HuaBio, ER1916-26, 1:500) primary antibodies. Then, the primary antibody was discarded and the sections were washed three times with PBS for 5 min each time. Incubate the sections with secondary antibodies for 1 hr and DAPI for 5 min, then wash with PBS. Seal the cell with glycerol and use fluorescent microscope to capture images.

## LDM nuclei isolation and 10x Genomics Chromium library and sequencing

LDM nuclei isolation and 10x Genomics Chromium library and sequencing were performed by LC-Bio Technology Co., Ltd. (Hangzhou, China) as per the previously published paper (*Wang et al., 2023b*). Briefly, nuclei of LDM samples were isolated, then homogenized and incubated for 5 min on ice. The homogenate was then filtered, centrifuged, and collected. The pellet was then resuspended, washed by the buffer, incubated and centrifuged. After centrifugation, the pellet was resuspended, filtered, and counted. Single-cell suspensions were loaded onto 10x Genomics Chromium for capturing 5000 single cells, followed by cDNA amplification, and library construction steps were performed. Libraries were sequenced using the Illumina NovaSeq 6000 sequencing system.

## Bioinformatics analysis

SnRNA-seq results were demultiplexed and converted to FASTQ format by using Illumina bcl2fastq software and followingly processed by the Cell Ranger. Then, the Seurat packages was used to analyze the Cell Ranger output. After the quality control, 22,540 cells were obtained. DoubletFinder package was used to remove doublets and Harmony package was used to perform batch correction of data integration between samples. We further used Seurat, UMAP, the FindAllMarkers function to visualize the data, find clusters, and select marker genes. Monocle 2 package was used to perform trajectory analysis and model differentiation trajectories. RNA velocity analysis was independently performed in FAPs by SAMTools and the Velocyto (*Bergen et al., 2020*). CellPhoneDB package was used for cell communication analysis and make further speculations about potential cellular interaction mechanisms.

## Primary FAPs isolation, magnetic cell sorting, and cell culture

Primary FAPs isolation were performed as previously described (*Wang et al., 2023b*). Briefly, a piece of muscle from a 3-day-old piglet was minced, and added five times the volume of 0.2% collagenase type I, then digested at 37°C for 1 hr. After filtering and centrifuging, add red blood cell lysate to split for 5 min at 4°C followed by incubation with a Dead Cell Removal Kit at room temperature for 15 min. Then, the CD140a antibody was added, followed by incubation, and centrifugation. Next, 20 µL antibiotin microbeads was added, followed by another incubation and centrifugation. After passing through the magnetic column, the cells on the adsorption column were PDGFRα⁺ cells. For FAPs' adipogenic differentiation, when the cells confluence reached 90%, the 10% FBS growth medium was replaced with induction medium after 4 days, the medium was changed to differentiation medium and cultured for another 4 days until the adipocytes were mature.

## Nile Red staining

Rinse cultured FAPs with 1× PBS three times, discard PBS, fix FAPs with 4% formaldehyde for 15 min, repeat the rinsing step, and then add Nile Red solution (1:500 for lipid droplet staining) and DAPI (1:500 for nuclei staining) for 5 min. Seal the cell with glycerol and use fluorescent microscope to capture images.

## Total RNA extraction and qPCR

Total RNA extraction and quantitative real-time PCR (qPCR) were conducted as described before (*Shan et al., 2016*). Briefly, total RNA of FAPs were extracted by using TRIzol, and the Spectro-photometer and a RevertAid First Strand cDNA Synthesis Kit were used to measure the purity and concentration of total RNA and reversed RNA samples. qPCR was performed by using Applied Biosystems StepOnePlus Real-Time PCR System with Hieff qPCR SYBR Green Master Mix and gene-specific primers (*Supplementary file 1*). Relative changes in gene expression were analyzed using the $2^{-\Delta\Delta CT}$ method and normalized using 18S ribosomal RNA as an internal control.

## Protein extraction and western blotting

Protein extraction and western blotting were carried out as mentioned previously (*Shan et al., 2016*). In brief, total proteins were isolated from cells or tissues with RIPA buffer. After measuring the concentrations, proteins were separated using SDS-PAGE and subsequently transferred to a polyvinylidene fluoride membrane (PVDF, Millipore Corporation). Then, block the PVDF membrane with blocking buffer (5% fat-free milk) for 1 hr and incubate with primary antibodies overnight at 4°C. The peroxisome proliferator-activated receptor γ (PPARγ) (C26H12, 1:1000) were purchased from Cell Signaling Technology (CST). The β-actin (M1210-2, 1:10000), FABP4 (E71703-98, 1:2000), SCD1 (ER1916-26, 1:1000), PDE4D (ER1916-26, 1:500), p-JNK1/2/3(T183+T183+T221) (ET1609-42, 1:2000), and JNK1/2/3 (ET1601-28, 1:2000) antibodies were from HuaBio. Dilution of the secondary antibody was 500-fold. The ChemiScope3500 Mini System was used for protein detection.

## Statistical analysis

GraphPad (Prism 8.3.0) was used for data analyses and R software (version 4.3.2) was used for data visualization. Data comparisons were made by unpaired two-tailed Student's t-tests and one-way ANOVA. Differences were considered significant at $p < 0.05$.

## Acknowledgements

We thank members of the Shan Laboratory for comments and this work was partially supported by the National Natural Science Foundation of China (32272887), the Natural Science Foundation of Zhejiang Province (LZ22C170003), and the 'Hundred Talents Program' funding from Zhejiang University to TZS.

## Additional information

### Funding

| Funder | Grant reference number | Author |
|---|---|---|
| National Natural Science Foundation of China | 32272887 | Tizhong Shan |
| Natural Science Foundation of Zhejiang Province | LZ22C170003 | Tizhong Shan |
| Zhejiang University | Hundred Talents Program | Tizhong Shan |

The funders had no role in study design, data collection and interpretation, or the decision to submit the work for publication.

## Author contributions
Liyi Wang, Conceptualization, Data curation, Software, Formal analysis, Validation, Investigation, Visualization, Methodology, Writing – original draft; Shiqi Liu, Investigation, Methodology, Writing – review and editing; Shu Zhang, Investigation; Yizhen Wang, Resources, Supervision; Yanbing Zhou, Conceptualization, Supervision, Investigation, Methodology; Tizhong Shan, Conceptualization, Resources, Supervision, Funding acquisition, Project administration, Writing – review and editing

## Author ORCIDs
Liyi Wang ⓘ https://orcid.org/0000-0002-9454-8872
Yanbing Zhou ⓘ https://orcid.org/0000-0002-4196-5947
Tizhong Shan ⓘ https://orcid.org/0000-0002-4738-414X

## Ethics
The Zhejiang University Animal Care and Use Committee approved all procedures and housing (ZJU20170466).

Joint public review: https://doi.org/10.7554/eLife.99790.4.sa1
Author response https://doi.org/10.7554/eLife.99790.4.sa2

# Additional files

## Supplementary files
Supplementary file 1. The primer sequence of quantitative real-time PCR (qPCR).

MDAR checklist

## Data availability
The raw snRNA-seq data reported in this paper have been deposited in the Genome Sequence Archive (*Chen et al., 2021*) in National Genomics Data Center (*CNCB-NGDC Members and Partners, 2024*), China National Center for Bioinformation / Beijing Institute of Genomics, Chinese Academy of Sciences (GSA: CRA022605), publicly accessible at https://ngdc.cncb.ac.cn/gsa.

The following dataset was generated:

| Author(s) | Year | Dataset title | Dataset URL | Database and Identifier |
|---|---|---|---|---|
| Wang L | 2025 | Single-nucleus RNA-seq of Heigai pigs | https://ngdc.cncb.ac.cn/gsa/browse/CRA022605 | Genome Sequence Archive, CRA022605 |

The following previously published dataset was used:

| Author(s) | Year | Dataset title | Dataset URL | Database and Identifier |
|---|---|---|---|---|
| Wang L, Zhao X, Liu S, You W, Huang Y, Zhou Y, Shan T | 2023 | Single-nucleus RNA-seq of pig | https://ngdc.cncb.ac.cn/gsa/browse/CRA011059 | Genome Sequence Archive, CRA011059 |

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
