## [Editor Report · eLife Assessment]

This study provides **valuable** information on the single nucleus RNA sequencing transcriptome, pathways, and cell types in pig skeletal muscle in response to conjugated linoleic acid (CLA) supplementation. Based on the comprehensive data analyses, the data are considered **compelling** and provide new insight into the mechanisms underlying intramuscular fat deposition and muscle fiber remodeling. The study contributes significantly to the understanding of nutritional strategies for fat infiltration in pig muscle.

---

## [Referee Report · Joint public review]

This study comprehensively presents data from single nuclei sequencing of Heigai pig skeletal muscle in response to conjugated linoleic acid supplementation. The authors identify changes in myofiber type and adipocyte subpopulations induced by linoleic acid at depth previously unobserved. The authors show that linoleic acid supplementation decreased the total myofiber count, specifically reducing type II muscle fiber types (IIB), myotendinous junctions, and neuromuscular junctions, whereas type I muscle fibers are increased. Moreover, the authors identify changes in adipocyte pools, specifically in a population marked by SCD1/DGAT2. To validate the skeletal muscle remodeling in response to linoleic acid supplementation, the authors compare transcriptomics data from Laiwu pigs, a model of high intramuscular fat, to Heigai pigs. The results verify changes in adipocyte subpopulations when pigs have higher intramuscular fat, either genetically or diet-induced. Targeted examination using cell-cell communication network analysis revealed associations with high intramuscular fat with fibro-adipogenic progenitors (FAPs). The authors then conclude that conjugated linoleic acid induces FAPs towards adipogenic commitment. Specifically, they show that linoleic acid stimulates FAPs to become SCD1/DGAT2+ adipocytes via JNK signaling. The authors conclude that their findings demonstrate the effects of conjugated linoleic acid on skeletal muscle fat formation in pigs, which could serve as a model for studying human skeletal muscle diseases.

[Editors' note: the authors have responded to the previous rounds of review: https://doi.org/10.7554/eLife.99790.1.sa1 and https://doi.org/10.7554/eLife.99790.2.sa1]

---

## [Author Response]

The following is the authors’ response to the previous reviews.

**Reviewer #1 (Public review):**
In this revised manuscript, the authors aim to elucidate the cytological mechanisms by which conjugated linoleic acids (CLAs) influence intramuscular fat deposition and muscle fiber transformation in pig models. They have utilized single-nucleus RNA sequencing (snRNA-seq) to explore the effects of CLA supplementation on cell populations, muscle fiber types, and adipocyte differentiation pathways in pig skeletal muscles. Notably, the authors have made significant efforts in addressing the previous concerns raised by the reviewers, clarifying key aspects of their methodology and data analysis.Strengths:(1) Thorough validation of key findings: The authors have addressed the need for further validation by including qPCR, immunofluorescence staining, and western blotting to verify changes in muscle fiber types and adipocyte populations, which strengthens their conclusions.(2) Improved figure presentation: The authors have enhanced figure quality, particularly for the Oil Red O and Nile Red staining images, which now better depict the organization of lipid droplets (Figure 7A). Statistical significance markers have also been clarified (Figure 7I and 7K).

Thanks!

Weaknesses:(1) Cross-species analysis and generalizability of the results: Although the authors could not perform a comparative analysis across species due to data limitations, they acknowledged this gap and focused on analyzing regulatory mechanisms specific to pigs. Their explanation is reasonable given the current availability of snRNA-seq datasets on muscle fat deposition in other human and mouse.

Thanks for your suggestion!

(2) Mechanistic depth in JNK signaling pathway: While the inclusion of additional experiments is a positive step, the exploration of the JNK signaling pathway could still benefit from deeper analysis of downstream transcriptional regulators. The current discussion acknowledges this limitation, but future studies should aim to address this gap fully.

Thanks! As we discussed in discussion part, further studies should focus on the downstream transcriptional regulators of JNK signaling pathway on IMF deposition.

(3) Limited exploration of other muscle groups: The authors did not expand their analysis to additional muscle groups, leaving some uncertainty regarding whether other muscle groups might respond differently to CLA supplementation. Further studies in this direction could enhance the understanding of muscle fiber dynamics across the organism.

Thanks for your suggestion! In this study, we mainly focused on the adipocytes, muscles and FAPs subpopulations, which play important roles in lipid deposition. As you suggested, our further study will focus on other subpopulations such as endothelial cells and immune cells.

**Reviewer #2 (Public review):**
Summary:This study comprehensively presents data from single nuclei sequencing of Heigai pig skeletal muscle in response to conjugated linoleic acid supplementation. The authors identify changes in myofiber type and adipocyte subpopulations induced by linoleic acid at depth previously unobserved. The authors show that linoleic acid supplementation decreased the total myofiber count, specifically reducing type II muscle fiber types (IIB), myotendinous junctions, and neuromuscular junctions, whereas type I muscle fibers are increased. Moreover, the authors identify changes in adipocyte pools, specifically in a population marked by SCD1/DGAT2. To validate the skeletal muscle remodeling in response to linoleic acid supplementation, the authors compare transcriptomics data from Laiwu pigs, a model of high intramuscular fat, to Heigai pigs. The results verify changes in adipocyte subpopulations when pigs have higher intramuscular fat, either genetically or diet-induced. Targeted examination using cell-cell communication network analysis revealed associations with high intramuscular fat with fibro-adipogenic progenitors (FAPs). The authors then conclude that conjugated linoleic acid induces FAPs towards adipogenic commitment. Specifically, they show that linoleic acid stimulates FAPs to become SCD1/DGAT2+ adipocytes via JNK signaling. The authors conclude that their findings demonstrate the effects of conjugated linoleic acid on skeletal muscle fat formation in pigs, which could serve as a model for studying human skeletal muscle diseases.Strengths:The comprehensive data analysis provides information on conjugated linoleic acid effects on pig skeletal muscle and organ function. The notion that linoleic acid induces skeletal muscle composition and fat accumulation is considered a strength and demonstrates the effect of dietary interactions on organ remodeling. This could have implications for the pig farming industry to promote muscle marbling. Additionally, these data may inform the remodeling of human skeletal muscle under dietary behaviors, such as elimination and supplementation diets and chronic overnutrition of nutrient-poor diets. However, the biggest strength resides in thorough data collection at the single nuclei level, which was extrapolated to other types of Chinese pigs.Weaknesses:Although the authors compiled a substantial and comprehensive dataset, the scope of cellular and molecular-level validation still needs to be expanded. For instance, the single nuclei data suggest changes in myofiber type after linoleic acid supplementation, but these findings need more thorough validation. Further histological and physiological assessments are necessary to address fiber types and oxidative potential. Similarly, the authors propose that linoleic acid alters adipocyte populations, FAPs, and preadipocytes; however, there are limited cellular and molecular analyses to confirm these findings. The identified JNK signaling pathways require additional follow-ups on the molecular mechanism or transcriptional regulation. However, these issues are discussed as potential areas for future exploration. While various individual studies have been conducted on mouse/human skeletal muscle and adipose tissues, these have only been briefly discussed, and further investigation is warranted. Additionally, the authors incorporate two pig models into their results, but they only examine one muscle group. Exploring whether other muscle groups respond similarly or differently to linoleic acid supplementation would be valuable. Furthermore, the authors should discuss how their results translate to human and pig nutrition, such as the desirability and cost-effectiveness for pig farmers and human diets high in linoleic acid. Notably, while the single nuclei data is comprehensive, there needs to be a statement on data deposition and code availability, allowing others access to these datasets.

Thanks for your suggestion!

**Recommendations for the authors:**

**Reviewer #2 (Recommendations for the authors):**
The authors have discussed and provided some experimental evidence to address the related issues to help justify their conclusions. The reviewer believes that authors should deposit their single-cell sequencing data and code for the broader research community.

Thank you! We have uploaded our raw dataset in the Genome Sequence Archive (Genomics, Proteomics & Bioinformatics 2021) in National Genomics Data Center (Nucleic Acids Res 2022), China National Center for Bioinformation / Beijing Institute of Genomics, Chinese Academy of Sciences and data availability part has been updated (line 575-579).